# Lanthanides Toxicity in Zebrafish Embryos Are Correlated to Their Atomic Number

**DOI:** 10.3390/toxics10060336

**Published:** 2022-06-19

**Authors:** Ying-Ting Lin, Rong-Xuan Liu, Gilbert Audira, Michael Edbert Suryanto, Marri Jmelou M. Roldan, Jiann-Shing Lee, Tzong-Rong Ger, Chung-Der Hsiao

**Affiliations:** 1Department of Biotechnology, College of Life Science, Kaohsiung Medical University, Kaohsiung 80708, Taiwan; ytlin@kmu.edu.tw (Y.-T.L.); fromoursound@gmail.com (R.-X.L.); 2Drug Development & Value Creation Research Center, Kaohsiung Medical University, Kaohsiung 80708, Taiwan; 3Department of Bioscience Technology, Chung Yuan Christian University, Taoyuan 320314, Taiwan; gilbertaudira@yahoo.com (G.A.); michael.edbert93@gmail.com (M.E.S.); 4Department of Chemistry, Chung Yuan Christian University, Taoyuan 320314, Taiwan; 5Faculty of Pharmacy, The Graduate School, University of Santo Tomas, Manila 1008, Philippines; mmroldan@ust.edu.ph; 6Department of Applied Physics, National Pingtung University, Pingtung 90003, Taiwan; 7Department of Biomedical Engineering, Chung Yuan Christian University, Taoyuan 320314, Taiwan; 8Research Center for Aquatic Toxicology and Pharmacology, Chung Yuan Christian University, Chung-Li 320314, Taiwan

**Keywords:** rare earth element, toxicity, zebrafish, atomic number, electronic structural factors

## Abstract

Rare earth elements (REEs) are critical metallic materials with a broad application in industry and biomedicine. The exponential increase in REEs utilization might elevate the toxicity to aquatic animals if they are released into the water due to uncareful handling. The specific objective of our study is to explore comprehensively the critical factor of a model Lanthanide complex electronic structures for the acute toxicity of REEs based on utilizing zebrafish as a model animal. Based on the 96 h LC_50_ test, we found that the majority of light REEs display lower LC_50_ values (4.19–25.17 ppm) than heavy REEs (10.30–41.83 ppm); indicating that they are atomic number dependent. Later, linear regression analyses further show that the average carbon charge on the aromatic ring (aromatic C_avg_ charge) can be the most significant electronic structural factor responsible for the Lanthanides’ toxicity in zebrafish embryos. Our results confirm a very strong correlation of LC_50_ to Lanthanide’s atomic numbers (r = 0.72), Milliken charge (r = 0.70), and aromatic C_avg_ charge (r = −0.85). This most significant correlation suggests a possible toxicity mechanism that the Lanthanide cation’s capability to stably bind to the aromatic ring on the residue of targeted proteins via a covalent chelating bond. Instead, the increasing ionic bond character can reduce REEs’ toxicity. In addition, Lanthanide toxicity was also evaluated by observing the disruption of photo motor response (PMR) activity in zebrafish embryos. Our study provides the first in vivo evidence to demonstrate the correlation between an atomic number of Lanthanide ions and the Lanthanide toxicity to zebrafish embryos.

## 1. Introduction 

Rare earth elements (REE) is a group of seventeen chemical elements consisting of Scandium (Sc), Yttrium (Y), and 15 lanthanide elements of Lanthanum (La), Cerium (Ce), Praseodymium (Pr), Neodymium (Nd), Promethium (Pm), Samarium (Sm), Europium (Eu), Gadolinium (Gd), Terbium (Tb), Dysprosium (Dy), Holmium (Ho), Erbium (Er), Thulium (Tm), Ytterbium (Yb), and Lutetium (Lu) [1,2,3]. They are additionally categorized into ‘light’ or ‘heavy’, depending on their relative atomic weights [4]. Because of their metallic nature, and similar characteristics, typically in oxide compounds, REEs have also resorted to rare earth metals [2].

Those REEs are metals and alloys which are used in several electronic devices in our daily lives [1]. Many portable electronic devices such as smartphones, laptops, cameras, and others require rechargeable REEs-containing batteries for continuous usage [5,6]. The REEs are also used as catalysts, phosphorous, and polishing compounds [7]. These REEs also have their importance for national security purposes. According to the United States Department of Defense, the military used night optical/observation devices, precision-guided weapons, military communications systems, GPS equipment, and other defense electronics made of REEs. Hard alloys made from these elements are used in armored personnel carriers and missiles [8,9].

Unfortunately, chronic exposure to REEs has been proven to cause an adverse effect on humans and other living organisms, including aquatic animals. The aquatic animals will suffer from the contaminated water source, and thus, the survival rate will be decreased [10]. The potential toxicity of REEs on aquatic animals has been extensively reviewed in our previous publication [10]. However, based on the review, most of the related studies of REEs toxicity to aquatic life forms were mostly confined to certain elements, leaving scarce information about the toxicity of other REEs and their colloids. Therefore, a comprehensive study on the toxicity of various REEs to aquatic animals is required to fill the existing information gap and establish an environment fingerprint for utilizing REEs.

This is because the REEs will disrupt brain neurotransmitters in humans and delay transmitting the neural signals, which are significant to ensure good communication between the brain and other parts of the body [11]. REEs also will affect animals’ health if they are exposed to the REEs frequently. The waste rock and dust from the mine may contaminate local soils, impacting the local wildlife respiratory system [12]. On the other hand, the waste from the mining of the REEs will leak into the river or pond if the wastage management is failed or is improper. The increase of contaminants and sediments will alter the water chemistry such as pH value, the concentration of oxygen, and carbon dioxide [13].

Toxicity is related to chemical nature, solubility, valence state, duration of exposure, and dose of the compound [14]. Factors such as the valence state and the nature of metal-binding ligands influence the chemical form. A previous study used density functional theory (DFT) to demonstrate that lanthanides’ electronic structure plays a vital role in REEs’ chemical and physical properties [15]. The general valence shell electronic configuration of the lanthanide series is *4f*^1−14^*6s*^2^. When present in the state of coordination complexes, most of the lanthanides are in the +3 oxidation state, Ln^3+^, while especially stable *4f* configurations can also give +4 (Ce, Tb) or +2 (Eu, Yb) ions [16]. All the former states are firmly electropositive. Therefore, lanthanide ions are classified as hard Lewis acids on account of the hard and soft acids and bases (HSAB) theory, as proposed by Pearson [17]. While the ionic radius of the lanthanides mainly sets on the +3-oxidation state (Ln^3+^), their ionic radii decrease steadily with the atomic number along with the series. Their *4f* orbitals show a poor shielding effect; therefore, there is a gradual increase in the effective nuclear positive charge experienced by the outer negative electrons. This regular decrease in the ionic radii is known as lanthanides contraction [18].

This result is generally recognized because of the subservience of the orbital radii with the total quantum number, which puts together the *4f* orbitals that are more adjacent to the nuclei than the *5d* and surpasses the *6s* orbitals. Therefore, the *6s* electrons and (customarily) one *4f* electron are lost, and the ions have the configuration [Xe]*4f*^0−14^ in Ln^3+^ compounds. Moreover, the *4f* electrons are effectively guarded against external agitation by the *5d* and *6s* shells, and as a result, they are not anticipated to be remarkably entangled in chemical bonding [19]. The probability that *4f* electrons exist in the outer regions of atoms or ions is very low, so there is almost no constructive overlap between the orbitals of the Ln^3+^ ion and any binding ligand, resulting in bonds with significant ionic character. Therefore, lanthanide complexes typically have slight or no covalent makeup and are not determined by orbital geometries. Thus, the lanthanides are different in their size because of their lack of orbital interactivity even though this deficiency does not affect their complexity. However, this presumption is a rough estimation, different spectroscopic characteristics of Ln^3+^ complexes signify a definite impact of the chemical environment on the *4f* electrons [20]. Following the DFT study on Ln(C_3_H_5_)Cp(OMe), a Lanthanide complex model, a prior study proposed the participation of *4f* orbitals in the bonding, opposing the broader perspective [15]. The Lanthanide, including *4f* electrons, chelates to propenyl, cyclopenta-1,3-dienyl, and methoxy ligands, respectively. They ascribed this rather an indirect effect: the quasi-degeneracy of the *4f* and *5d* orbitals in the complexes. Given the various theoretical studies conducted on Ln^3+^ complexes, we believe that *4f* electrons have a limited role (if any) in the chemical bonding of most systems, especially for coordination compounds. As stated above, it is commonly accepted that the lanthanide-ligand relationships are anticipated to be principally electrostatic, which brings on diversification of coordination numbers and the absence of preset chemical bonds directionality [21]. Although toxicity cannot be directly related to the metal complex structure, toxicity may be related to the stability and solubility of the complexes in a test system.

The foremost advantage of using zebrafish as a toxicological model is that they are small compared to other fish species. The adult zebrafish are only about 1–1.15 inches in length, which considerably minimizes housing space and expenses [22,23]. Furthermore, the life cycle of zebrafish is short, which only takes approximately 90 days from an egg to an adult and thus, the result can be obtained in a short period [24]. Furthermore, besides their size and short life cycle, this species has high fecundity which allowed for high-throughput screens for toxicity testing. Adult female zebrafish can lay 200–300 eggs in a single breeding process every 5–7 days [23]. In addition, their nature permits for uncomplicated development staging and growth tracking during the embryo stage. It is known that the zebrafish have stage-specific patterns of motility or locomotion [25]. These allow observation of the locomotion changes of fishes toward the chemical stimuli. For example, the toxicities of the compounds can be observed by disruption or alteration in zebrafish motoric responses [26]. Zebrafish make an excellent model organism due to their small size, inexpensive, and easy husbandry which is followed by rapid developmental toxicity assays such as lethal concentration for 50% (LC_50_) and locomotor assay [27]. Taken together, this experiment aimed to investigate the possible correlation between electronic structures of lanthanide complexes and REEs toxicity in zebrafish embryos.

## 2. Materials and Methods

### 2.1. Zebrafish Maintenance

AB strain zebrafish originated from Taiwan Zebrafish Core Facility at Academia Sinica (TZCAS) and had been reared in the zebrafish facility for approximately two years. The zebrafish stock was kept in the laboratory in a recirculating water system at 27 ± 1 °C with a 14/10 light and dark cycle shift. The water system with 0.254 ± 0.004 mS/cm of electrical conductivity, 6.5 ± 0.2 mg/L dissolved oxygen, 183 ± 5 mg of CaCO_3_/L for water hardness and alkalinity, and pH 7.2 ± 0.4, was sterilized by UV light. A biofilter was used to maintain ammonia levels <0.1 mg/L, nitrite (NO^2−^) < 0.2 mg/L, and nitrate (NO^3−^) <20 mg/L. The fish were fed twice a day with either commercial dry food or lab-grown brine shrimp. For the breeding process, the fishes with a ratio of 1:2 (one female and two males) according to previously published studies [28,29] were transferred into a breeding tank with a glass separator between them on the night before the breeding process. On the next morning, the separator was removed to initiate the mating process. Afterward, the embryos were collected and transferred in methylene blue water. To maintain the embryo’s environment quality, all of the dead embryos were removed on the following day, and methylene blue water was changed. All these animal experimental protocols and ethics were approved by the Chung Yuan Christian University animal care and welfare committee (Number: CYCU109001, issue date 20 January 2020).

### 2.2. Chemical Preparation

Later, we collected 43 electronic structural parameters from a density functional theory (DFT) computation of the literature and Lanthanide’s atomic number. Finally, a total of 44 physical property endpoints for a given Lanthanide can be used as an input for correlation studies (Appendix A). Seventeen REEs of Gadolinium chloride hexahydrate (GdCl_3_·6H_2_O), Yttrium chloride hexahydrate (YCl_3_·6H_2_O), Lanthanum chloride, hexahydrate (LaCl_3_·6H_2_O), Cerium(III) chloride heptahydrate (CeCl_3_·7H_2_O), Samarium chloride hexahydrate (SmCl_3_·6H_2_O), Holmium chloride hexahydrate (HoCl_3_·6H_2_O), Dysprosium chloride hexahydrate (DyCl_3_·6H2O), Neodymium chloride hexahydrate (NdCl_3_·6H_2_O), Europium chloride hexahydrate (EuCl_3_·6H_2_O), Terbium chloride hexahydrate (TbCl_3_·6H_2_O), Praseodymium chloride (PrCl_3_), Erbium chloride hexahydrate (ErCl_3_·6H_2_O), Ytterbium chloride (YbCl_3_), Lutetium chloride hexahydrate (LuCl_3_·6H_2_O), Thulium (III) chloride hexahydrate (TmCl_3_·6H_2_O), Scandium chloride (ScCl_3_), Terbium chloride hexahydrate (TbCl_3_·6H_2_O) were purchased from Aladdin (Shanghai, China). The REE powders were dissolved in double-distilled water (ddH_2_O) in a stock solution at a 1000 ppm concentration. Due to the radioactivity nature of Promethium (Pr), this chemical was not used for toxicity tests in our study.

### 2.3. Zebrafish Acute Toxicity Test (LC_50_)

For zebrafish toxicity assessment, we conducted 96 h lethal concentration of 50% (LC_50_), which is the concentration to kill 50% of the fish, for each Lanthanide to obtain the 96 h LC_50_ value. Acute toxicity tests in zebrafish larvae were done according to OECD Guidelines Section 2 No. 203 (https://www.oecd-ilibrary.org/environment/test-no-203-fish-acute-toxicity-test_9789264069961-en, accessed on 12 December 2021). The test was also based on several prior studies on REEs toxicities in several aquatic animal models [30,31,32]. After the calculations, serial dilutions of REEs from the stock solution with methylene blue water were conducted to achieve the desired test concentrations, which were 0 (control), 0.01, 0.1, 1, 10, and 100 ppm. Even though these concentrations were relatively higher than REE concentrations found in the surface water measured in the previous finding [33], they were chosen since several prior studies in REEs demonstrated the LC_50_ values of these metals in various aquatic organisms are belong to the current concentration range [31,34]. In addition, these concentrations were also determined according to OECD guidelines. Prior to the exposure, the mixtures were homogenized. Afterward, twenty zebrafish eggs aged 24 h post-fertilization (hpf) were placed into a 3.5 cm petri dish with ~10 mL of the solutions. During exposure, petri dishes were transferred into the incubator at 26 °C under a 10/14-h dark/light regime. The mortality rate was documented every 24 h at 48, 72, and 96 hpf, and dead embryos or larvae were removed at every examination. Unfortunately, due to the equipment limitation, the quantification of the REEs in the test solutions was not available in the current study. However, this limitation does not neglect the toxicity results obtained from the experiment since this study also focused in comparing the toxicity of each Lanthanide in the same exposure condition. The 96 h LC_50_ values were calculated by using the sigmoid curve fitting method provided by GraphPad Prism software (GraphPad Prism 8.0.2, GraphPad Software, Inc., San Diego, CA, USA). This experiment was carried out in two replicates.

### 2.4. Zebrafish Locomotor Assay and Photo Motor Response (PMR) Test

Next, a locomotion test was also carried out to conduct a zebrafish toxicity assessment for each Lanthanide by calculating several behavior endpoints, which were total distance traveled, total burst count, and rotation count in both light and dark cycles. Then, the photo motor response (PMR) of zebrafish toward REE exposure was calculated. PMR test is a method to evaluate how zebrafish embryos respond to light and dark stimuli by measuring three important endpoints of total distance traveled, rotation count, and burst count. The last and first minute of each photoperiod (three dark and three light cycles) are used to calculate PMRs [35]. We followed our previously published protocol with some modifications to conduct the PMR test using the ZebraBox instrument and build-in software (Viewpoint, France) [36]. Zebrafish embryos aged at 96 hpf were incubated with REEs at 1 ppm concentration for 24 h in a 9 cm diameter Petri dish, and later individually transferred to 48 well plates for PMR measurement at 120 hpf.

### 2.5. Electronic Structures of Lanthanide Complexes

Forty-five electronic structural parameters, plus with Lanthanide’s atomic number, were taken from the literature [15], which stored the density functional theory (DFT) calculation of the fourteen Lanthanide complexes [Ln(C_3_H_5_)Cp(OMe)] (Ln = La-Lu, C_3_H_5_ = propenyl, Cp = cyclopenta-1,3-dienyl, C_5_H_5_, OMe = methoxy). The Lanthanide complexes’ electronic structural parameters were calculated by Schinzel and colleagues using the Kohn-Sham methodology [37] in the DFT with the ADF03 and ADF04 program packages. DFT is one of the sophisticated calculating methodologies of quantum chemistry [20]. Various types of electronic structural parameters used include bond length, Mulliken charge, Hirshfeld charge, seven kinds of energies (HOMO (Highest Occupied Molecular Orbital), LUMO (Lowest Unoccupied Molecular Orbital), electrostatic, kinetic, Coulombic, exchange-correlation, and total bonding), and Mulliken populations in valence orbitals (see Appendix A).

### 2.6. Statistics: Lanthanides’ Electronic Structures Correlates with REEs’ Toxicity

All electronic structural parameters of Lanthanides have been comprehensively correlated to the experimental end, the lethal concentration 50% (LC_50_) by the linear regression analyses. Statistical analyses were done using R (Available online: https://www.r-project.org, accessed on 24 January 2022). Graphics were done using the GraphPad Prism 8 software (Available online: https://www.graphpad.com/scientific-software/prism/, accessed on 24 January 2022). The statistical significance for heavy REE (HREE) and light REE (LREE) comparison was analyzed using a one-tailed Student’s *t*-test with a significant level setting at *p* < 0.05, resulting in the *p*-values, 0.0111. Two sets of data variables have shown normal distributions (Gaussian form) with the *p*-value of 0.1824 from the Kolmogorov-Smirnov normality test, and homogeneity of variance with the result of the F-test (F = 2.442).

## 3. Results

The 96 h LC_50_ values of Lanthanides demonstrated their toxicity is related to their relative atomic weights (Figure 1). The LC_50_ level was measured and predicted by a sigmoid curve fitting method after the zebrafish embryos were treated with different concentrations of Lanthanide (10^−2^ to 10^2^ ppm). It displayed those smaller atomic weights (LREE) were more toxic than the larger sized ones (HREE). Most of the LREE have lower LC50 values which were in the range of 4.19–25.17 ppm (Figure 1A) compared to HREE which in the range of 10.30–41.83 ppm (Figure 1B).

For the locomotor assay, the experimental data are presented as the mean values and the mean deviation values (from control) of the distance and counts for zebrafish locomotion. The deviation value is the mean experimental value minus the control. For example, any specific dark and light cycle total burst count is subtracted from the control count. The swimming activity chronology (total 80 min) of zebrafish following 24 h exposure to 1 ppm Lanthanide is presented in Figure 2. Both LREE and HREE Lanthanides could induce alteration in one or more zebrafish locomotion endpoints. From the LREE group, La increased the average burst count (Figure 2C), meanwhile, Eu could increase the activity in all behavior endpoints tested (distance, burst, and rotation) (Figure 2A,C,E). Meanwhile, from the HREE group, Yb and Lu constantly reduced all the three endpoints tested (Figure 2B,D,F). Another HREE, Ho and Er increased the total distance activity (Figure 2B) but reduce the burst count (Figure 2D). Based on the rotation count endpoint, Nd, Sm, and Eu from the LREE group (Figure 2E) and Er from the HREE group (Figure 2F) have higher rotation activity compared to the control. The further statistical significance details are presented in Appendix A.

In Figure 3, several Lanthanides significantly affected the PMR of zebrafish which is displayed by the changes in distance traveled movement between dark to light or light to dark transition. Three dark and three light periods of photomotor response are measured. The last and first minute of each photoperiod from the total distance endpoint are used to calculate PMR. From the LREE group, two Lanthanides: Ce and Pr disrupted the PMR activity by significantly increased the movement in light conditions (Figure 3A). Meanwhile, in the HREE group, Tb, Dy, Er, Tm, Yb, and Lu significantly altered the PMR with decreased in dark and increased in light conditions (Figure 3B).

After conducting the Pearson correlation test systematically, we found 96 h LC_50_ has significant correlations (r > 0.7) with the three critical parameters, i.e., atomic number (A.N.), Lanthanide’s Milliken charge, and the average Milliken charge of the aromatic ring carbons (Aromatic C_avg_ Charge). The density functional theory (DFT) calculating values for these three parameters are listed in Table 1. The other endpoints coming from the PMR test, however, display low Pearson’s correlation coefficient (|r|) (less than 0.3) with low average deviation values (|r| < 0.5).

In Figure 4A, the Pearson correlation analysis showed a significant correlation between LC_50_ and Lanthanide’s atomic numbers with r = 0.72. The lanthanide is getting less toxic (i.e., higher LC_50_) with the increase of atomic number, suggesting that the majority of LREEs (La to Gd) are more toxic than HREEs (Tb to Lu) in zebrafish embryos. In Figure 4B, a violin plot shows the significant difference for LC_50_ between LREEs and HREEs (*p*-value = 0.0111), supporting that most of the tested LREEs (La to Gd) are more toxic than HREEs (Tb to Lu). In Figure 4C, the Pearson correlation analysis showed another significant correlation between LC_50_ and Lanthanide’s Milliken charge with r = 0.70. The less positive Milliken charge of Lanthanide showed more toxicity for specific REEs. In Figure 4D, the correlation analysis showed the most significant negative correlation between LC_50_ and the aromatic C_avg_ charge with r = −0.85. The less negative value of the aromatic C_avg_ charge display more toxicity for a specific REE. A summary of the above observations leads to a significant finding that the positive Lanthanide charge and the negative aromatic C_avg_ charge can be the most significant electronic structural factors responsible for REEs’ toxicity in zebrafish. However, even though most of LREEs displayed higher toxicity than HREEs, one has to keep in mind that in the present study, one of the LREEs, which was Gd, was less toxic than some HREEs.

In Figure 5, we show that the property of the chelating bond of Lanthanide (III) cation to chelate aromatic ring is lying between covalent and ionic character (covalency/ ionicity). In the DFT calculation of the model Lanthanide complexes, Ln (III) can chelate Cp (cyclopentadienyl) by inducing an average Milliken charge of the carbons on the Cp ring (aromatic C_avg_ charge). Here, La (III) complex’s Cp is shown as an example. This aromatic C_avg_ charge with −0.432 e of Milliken charge will increase along with the Lanthanide series until Lu (III) complex’s Cp charge, −0.460 e, indicating that the Cp becomes more ionic. Similarly, a La (III) with +1.900 e of cation’s charge is increased along with the Lanthanide series until Lu (III) cation’s charge is +2.254 e, which indicates that Lu cation itself is the most ionic.

## 4. Discussion

By systematically evaluating REEs’ quantum chemistry, we found a solid correlation between REEs toxicity on zebrafish embryos (based on LC_50_) with REEs’ electronic structures, which are Lanthanide’s atomic numbers (r = 0.72), Lanthanide’s Milliken charge (r = 0.70), and aromatic C_avg_ charge (r = −0.85) for the first time. From the results, the LC_50_ assessment demonstrates that the majority of LREEs displayed higher toxicity than HREEs in zebrafish embryos. This result is similar to the previous study that demonstrated the two properties of REEs, which were atomic number and ionic radius, could modulate REEs lethal toxicity (LC_50_ and EC_50_) in *Hydra attenuate* [34]. Surprisingly, based on locomotion and PMR data, the majority HREEs showed more disruption in zebrafish movement than LREE. Contrary to the current LC_50_ results in the aquatic environment, a previous study demonstrated that the toxicity of HREEs is relatively higher than LREEs in soils [38]. Meanwhile, another experiment showed that lanthanides’ toxicity to *Hyalella azteca* was decreased from La to Er, but reversed for the three heaviest elements (Tm, Yb, Lu) [39]. Regarding this matter, a study comparing the lethal effects and PMR effects in zebrafish embryos has been reported. Ortmann et al. (2020) displayed the effective concentrations of various neurotoxic substances to induce the decreased and/or increased movement activity were much lower than their LC_50_ concentrations [40]. Another study also demonstrated that PMR effect concentrations are well below baseline lethal toxicity [41]. This might indicate a possibility that behavioral testing is more sensitive than lethality measures. It is also reported that PMR effects are concentration-dependencies [40]. This may also explain the relative difference between the PMR effect and LC_50_ concentrations observed in this study. Since, only a single concentration (1 ppm) was used in the PMR assay, thus, the correlation of PMR and LC_50_ test at this stage is still preliminary. A prior experiment also indicated that HREEs, such as Lu, Eu, Tb, Ho, and Tm, have higher toxicity coefficients than LREEs [42]. The toxicity coefficient is a commonly used potential ecological risk index that is determined by elemental abundance and release effect [43]. However, the toxicity coefficients of REEs might be not parallel to the toxicity response coefficients, which also need the calculation of the metal bio-production index that is mainly based on the sensitivity of water to metal elements, for all model organisms [42]. In addition, the current finding also demonstrated that Gd, one of the LREEs, possessed less toxicity compared to some HREEs, which need further investigation in future studies. The difference in the toxicity rank of each element from several studies showed that the experimental results depend on the toxicological endpoint and test organism, and unfortunately, no single theory is capable of accommodating even most of the available experimental data regardless of the endpoints [44]. Therefore, whether HREEs are more toxic than LREEs (or vice versa) is still an unanswered question due to the lack of systematic comparison. Nevertheless, it is believed that the free ion is the most active metal form that living organisms can use. Considering the ionic character described above, a stronger capability to chelate aromatic rings can be inferred for HREEs. As a result, more LREE ions will be released in the aqueous solution than HREE ions for aqueous solutions containing the same concentration of Ln(C_3_H_5_)Cp(OMe). As stated previously, the lanthanide is getting less toxic (i.e., higher LC_50_) to zebrafish embryos with the increase of atomic number. Therefore, we suggested that the higher toxicity of LREEs observed in the current study might be related to their high solubility properties that make them easier to be exposed to zebrafish embryos compared to HREEs, which compensate for the absence of analytical measurements in the current study. This assumption is based on a prior study on the acute toxicity (LC_50_) of metals toward zebrafish larvae. From this study, several metals (Pb and Zn) resulted in mortalities since their solubility was increased after the exposures were conducted in soft water. Meanwhile, this toxicity did not occur when the exposures were carried out in the hard water, which most likely stemming from low solubility [45]. In addition, our study also provides a good in vivo backup support for the hypothesis proposed in a previous publication by using the DFT method [15].

The HSAB theory is a useful tool to predict the thermodynamic stability or kinetic lability of metal-ligand complexes. Based on the large electronegativity differences between hard acids and hard bases, hard acids prefer to bind with hard bases to give ionic complexes with high charge density that are not very polarizable but able to form bonding interactions that are more ionic in nature [17,46,47]. In the case of lanthanides, their ions exist overwhelmingly in the +3 oxidation state, which makes them a hard Lewis acid due to their high positive charge and small radius, forming strong bonds that cause a low solubility in water. As described in Table 1, the calculated ionic charges of both Ln^3+^ and aromatic C_avg_ tend to increase with an increasing atomic number of lanthanide series. Therefore, it is clear that the ionic character is more profound for the Ln(C_3_H_5_)Cp(OMe) complexes of late lanthanides due to the lanthanide contraction. Accordingly, the stability of the Ln(C_3_H_5_)Cp(OMe) complexes roughly increases with an increasing atomic number of the lanthanides. A related study also indicated that the stability constants of the late lanthanide complexes are larger than those of the early ones [48]. In addition, another study also reported that the larger and lighter REEs are less soluble ions while the smaller and heavier REEs are more soluble ions [49]. Altogether, normally, the light lanthanide complexes with weaker ionic character are easier to be dissociated into their constituent ions when they are dissolved in water than the heavy ones. In the future, the extensive usage of LREEs might become a major concern since they can cause bioconcentration, an accumulation of waterborne chemicals by aquatic animals through non-dietary routes, by delivery to the absorbing epithelium followed by movement across diffusion barriers into the blood [50]. Actually, the bioaccumulation of several REEs had been demonstrated by several prior studies in several models, including zooplankton and rainbow trout [51,52]. Moreover, in severe cases, bioconcentration leads to biomagnification, as the chemicals are already accumulated via the food chain following various pathways along with different trophic levels [53].

Quantum chemistry methods (in this case, DFT) can be used to calculate any atom’s partial charge in a molecule, of which one of them is being called the atom’s Milliken charge, which describes the final state of any chelating ligand to Ln^3+^. The model complex, Ln(C_3_H_5_)Cp(OMe), shows three typical types of organic moieties to chelate Ln^3+^, aromatics, alkenes, and oxygens, respectively. Various biological molecules in the organism body (zebrafish embryo) and different organic matter in natural waters can chelate Ln^3+^ by these three organic moieties. The chelating bond’s property for Lanthanide elements lies between the covalent and ionic character in the electronic structural aspect. The degree of covalency/ionicity in Lanthanide-carbon bonds may be a major possible electronic structural factor related to the acute toxicity of zebrafish embryos. As shown in Figure 1, the more the positive charge, the stronger the ionic character of a Lanthanide. The ideal ionic charge, i.e., formal charge, is +3 e if Lanthanide is totally ionized. Moreover, on the contrary, the more negative aromatic C_avg_ charge along the Lanthanide series displays a stronger ionic character for the aromatic ring. In the Lanthanide complex model, the most ionic Cp charge would be the value, −6/5 e. If it is possible to be fully ionized, five Cp hydrogen electrons contribute to the aromatic system. However, this complete ionization of Lanthanide complexes is not possible to occur because the ionized Lanthanide elements and chelated ligands will be entirely dissolved in a solvent. Therefore, a reasonable conjecture may be made that Lanthanide’s capability to chelate aromatic rings stably is crucial for the acute toxicity of zebrafish embryos. However, too much negative charge with increasing ionic character along the Lanthanide series might reduce their chelating ability to their target proteins and reduce such toxicity.

In conclusion, this study provides the first comprehensive toxicity assessment of 16 REEs at the whole organism level in zebrafish based on the LC_50_ acute toxicity test for the first time. Several interesting correlations were found between LC_50_ vs. A.N., LC_50_ vs. Ln (III) charge, and LC_50_ vs. aromatic C_avg_ charge. Those results demonstrated that REEs acute toxicity in zebrafish embryos is well correlated with the electronic structures of lanthanide complexes. However, ones have to keep in mind that the present ecotoxicity experiment still faced limitations, including the minimum analytical verification of exposure concentration, even though several actions had been taken to maintain the concentration of the compounds. Finally, based on the current findings, future studies are still needed in evaluating the toxicity of REEs toward aquatic organisms in different endpoints and other perspectives besides mortality-based endpoints to provide better insights into REE’s toxicity. Nevertheless, considering the increasing usage of these metals in modern technology, the current findings highlight the potential impact of REEs in freshwater systems, especially LREEs since they are more abundant than the heavier ones, which might be originated from their disposal in landfills, where leachates might start affecting this system and terrestrial system [34].

## Figures and Tables

**Figure 1 toxics-10-00336-f001:**
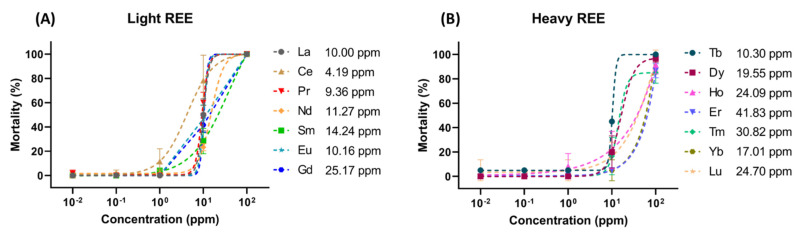
LC_50_ of zebrafish embryos exposed to actual concentrations (ppm) of (**A**) Lanthanide’s light REE and (**B**) Lanthanide’s heavy REE for 96 h. The 96 h LC50 values were calculated by using the sigmoid curve fitting method and the data are expressed as mean ± SEM.

**Figure 2 toxics-10-00336-f002:**
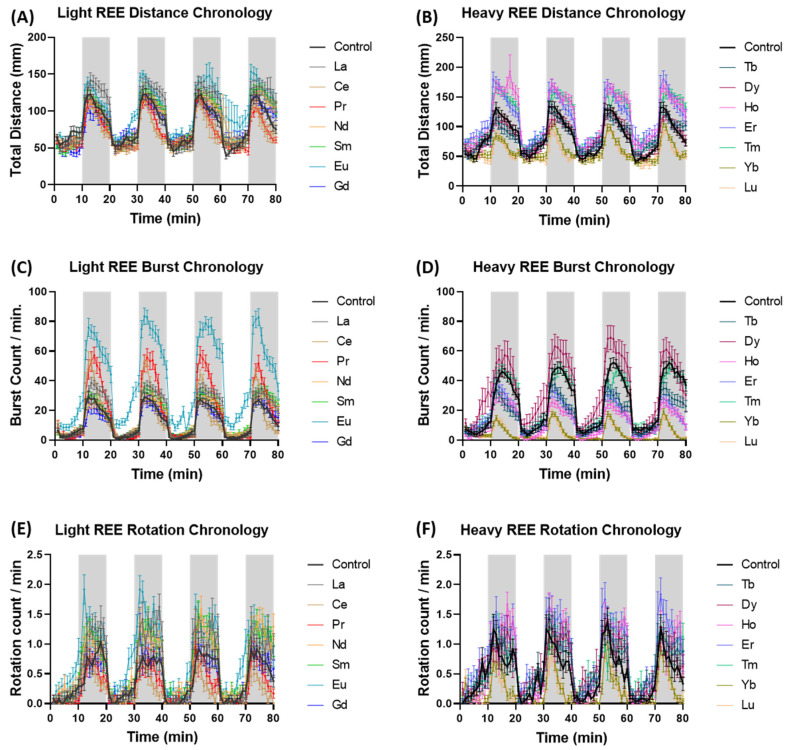
Swimming activity of zebrafish after exposure to different Lanthanides. The mean ± SEM of (**A**,**B**) distance traveled, (**C**,**D**) burst movement, and (**E**,**F**) rotation counts per minute by zebrafish larvae after 24 h exposure of 1 ppm LREE and HREE during both light and dark cycles. A two-way ANOVA test with Geisser-Greenhouse’s correction continued with Dunnett’s multiple comparisons test was carried out to compare all treatments with the control (presented in Appendix A).

**Figure 3 toxics-10-00336-f003:**
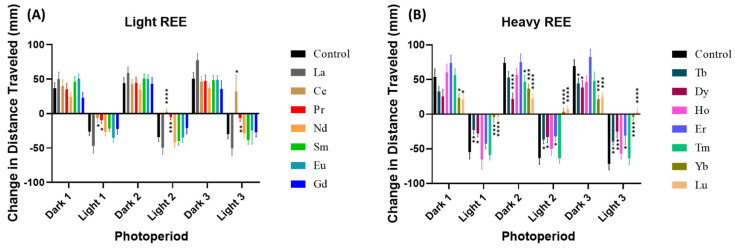
The photo motor response (PMR) of zebrafish after 96 h exposure to 1 ppm of (**A**) light REE and (**B**) heavy REE. The data are expressed as mean ± SEM and statistical differences were analyzed by two-way ANOVA followed with Fisher’s LSD test (* *p* < 0.05, ** *p* < 0.01, *** *p* < 0.001, **** *p* < 0.0001).

**Figure 4 toxics-10-00336-f004:**
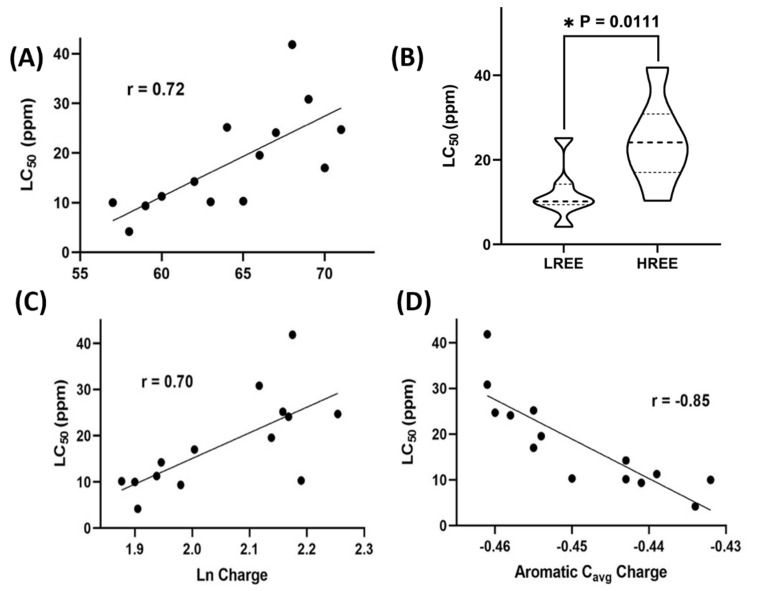
Correlations of properties of the Lanthanides complexes to the acute toxicity of fourteen Lanthanides. (**A**) The linear regression analysis showed a significant correlation between LC_50_ and Lanthanide’s atomic numbers with r = 0.72. (**B**) A violin plot shows a statistical difference between the two LC_50_ of LREE and HREE (*p* = 0.0111). (**C**) The linear regression analysis showed a significant correlation between LC_50_ and Lanthanide’s Milliken charge with r = 0.70. (**D**) The linear regression analysis showed the most significant correlation between LC_50_ and the aromatic C_avg_ charge with r = −0.85.

**Figure 5 toxics-10-00336-f005:**
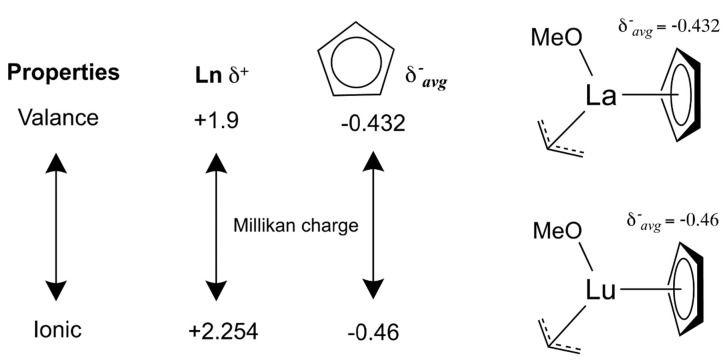
Summary of the covalency and ionicity property of the chelating bonds of Lanthanide (III) complexes.

**Table 1 toxics-10-00336-t001:** Lists are LC_50_, Lanthanide’s atomic number, an index for heavy rare earth elements, Lanthanide’s charge, and aromatic C_avg_ charge.

REEs	LC_50_ (ppm)	Atomic Number	Valence Electron	HREE or LREE	Ln Milliken Charge	Aromatic C_avg_ Charge
La	10	57	*5d* ^1^ *6s* ^2^	LREE	1.900	−0.432
Ce	4.187	58	*4f* ^1^ *5d* ^1^ *6s* ^2^	LREE	1.905	−0.434
Pr	9.36	59	*4f* ^3^ *6s* ^2^	LREE	1.980	−0.441
Nd	11.27	60	*4f* ^4^ *6s* ^2^	LREE	1.938	−0.439
Sm	14.24	62	*4f* ^6^ *6s* ^2^	LREE	1.946	−0.443
Eu	10.16	63	*4f* ^7^ *6s* ^2^	LREE	1.877	−0.443
Gd	25.17	64	*4f* ^7^ *5d* ^1^ *6s* ^2^	LREE	2.158	−0.455
Tb	10.3	65	*4f* ^9^ *6s* ^2^	HREE	2.190	−0.45
Dy	19.55	66	*4f* ^10^ *6s* ^2^	HREE	2.138	−0.454
Ho	24.09	67	*4f* ^11^ *6s* ^2^	HREE	2.168	−0.458
Er	41.83	68	*4f* ^12^ *6s* ^2^	HREE	2.175	−0.461
Tm	30.82	69	*4f* ^13^ *6s* ^2^	HREE	2.117	−0.461
Yb	17.01	70	*4f* ^14^ *6s* ^2^	HREE	2.004	−0.455
Lu	24.7	71	*4f* ^14^ *5d* ^1^ *6s* ^2^	HREE	2.254	−0.460

Note that HREE: heavy rare earth element. LREE: light rare earth element. Ln Milliken Charge: the calculated Milliken charge of Lanthanide when Lanthanide forms the model complex; Aromatic C_avg_ Charge: the average Milliken charge of the carbons on the aromatic ring in the Ln complex. The density functional theory (DFT), a quantum chemistry methodology, calculated both of the charges based on a series of Lanthanide chelated complex models, [Ln(C_3_H_5_)Cp(OMe)]. (Ln = La-Lu, C_3_H_5_ = propenyl, Cp = cyclopenta-1,3-dienyl, C_5_H_5_, OMe = methoxy).

## Data Availability

The data presented in this study are available direct to the corresponding authors.

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
