# Peer review of "Lanthanides Toxicity in Zebrafish Embryos Are Correlated to Their Atomic Number"

_toxics, 2022, doi:10.3390/toxics10060336_

Round 1
Reviewer 1 Report
I think a general light rewrite for English is appropriate. Some things just look incorrect. For instance, in Materials and Methods section under Zebrafish maintenance, "AB strain zebrafish was originated from the Taiwan..". I think that was should be eliminated for proper English. This occurs in a couple places in the manuscript and needs to be fixed.
As well, in the same paragraph, there is a statement that may not have been meant and could be problematic. The statement that zebrafish strain "has been reared in the zebrafish facility for more than 20 generations" . This may sound picky, but I'm sure you don't mean 20 successive generations referring to the animals that you were using. This would be problematic due to introduction of mutations, and poor health overall, particularly reproductive health. I know this wasn't what was meant in the statement. A clarification here. Perhaps it is better stated "(TZACS) where AB zebrafish have been reared for (fill in number ) years." This would avoid the misconception.
Author Response
Comments and Suggestions for Authors
I think a general light rewrite for English is appropriate. Some things just look incorrect. For instance, in Materials and Methods section under Zebrafish maintenance, "AB strain zebrafish was originated from the Taiwan.". I think that was should be eliminated for proper English. This occurs in a couple places in the manuscript and needs to be fixed.
Thank you so much for this suggestion. English editing and spell check have been made. In addition to the above comments, spelling and grammatical errors pointed out by the reviewers have been corrected.
As well, in the same paragraph, there is a statement that may not have been meant and could be problematic. The statement that zebrafish strain "has been reared in the zebrafish facility for more than 20 generations" . This may sound picky, but I'm sure you don't mean 20 successive generations referring to the animals that you were using. This would be problematic due to introduction of mutations, and poor health overall, particularly reproductive health. I know this wasn't what was meant in the statement. A clarification here. Perhaps it is better stated "(TZACS) where AB zebrafish have been reared for (fill in number ) years." This would avoid the misconception.
Thank you for pointing this out. The authors agree with this comment. We have accordingly revised the sentence to avoid the misconception as the reviewer’s suggestion. Therefore, the sentence in line 130-131 was revised to “AB strain zebrafish originated from Taiwan Zebrafish Core Facility at Academia Sinica (TZCAS) and had been reared in the zebrafish facility for approximately 2 years.
Reviewer 2 Report
Dear Authors,
The manuscript is interesting and well written. I have only two minor comments:
1) abstract should be changed; now it is too general; numerical values should be included
2) In M&M section concentrations of ammonia, nitrites and nitrates should be added (it is very important in toxicological studies)
Author Response
Comments and Suggestions for Authors
Dear Authors,
The manuscript is interesting and well written. I have only two minor comments:
1) abstract should be changed; now it is too general; numerical values should be included
Thank you so much for your suggestion. Therefore, the abstract has been updated with additional numerical values, such as LC50 values and their correlation to the Lanthanide’s atomic number, Milliken charge, and aromatic charge.
2) In M&M section concentrations of ammonia, nitrites and nitrates should be added (it is very important in toxicological studies)
The authors thank the reviewer for pointing this out. The Material and Methods section (2.1 Zebrafish maintenance) has been updated with the specific condition of ammonia, nitrite, and nitrate levels. In line 135-137, a sentence “A biofilter was used to maintain ammonia levels <0.1 mg/L, nitrite (NO2-) <0.2 mg/L, and nitrate (NO3-) <20 mg/L” was added.
Reviewer 3 Report
Ying-Ting Lin et al. study was aimed at demonstrating the lanthanides toxicity against zebrafish embryo and show that this toxicity decreases with atomic number.
Unfortunately in this form the manuscript can not be taken under consideration for the publication. The results are absolutely not relevant to the title. The Authors did the LC50 test. I find it necessary to present the results from this test – images illustrating the malformations, tables with scores, e.g. mortality (GraphPad prism enables creating very informative survival curves).
The results from PMR test are weakly and illogically presented. It is difficult for the reader to understand what exactly resulted from this tests. Maybe it would be a good idea to include some results from supplementary materials into the main text.
Author Response
Comments and Suggestions for Authors
Ying-Ting Lin et al. study was aimed at demonstrating the lanthanides toxicity against zebrafish embryo and show that this toxicity decreases with atomic number.
Unfortunately in this form the manuscript can not be taken under consideration for the publication. The results are absolutely not relevant to the title. The Authors did the LC50 test. I find it necessary to present the results from this test – images illustrating the malformations, tables with scores, e.g. mortality (GraphPad prism enables creating very informative survival curves).
Thank you so much for your concerns. However, regarding LC50 test, the authors did not capture the images that illustrating the malformations. As stated in line 178-182: “twenty zebrafish eggs aged 24 hours post-fertilization (hpf) were placed into a 3.5 cm petri dish with ~10 ml of the solutions. The mortality rate was documented every 24 hours at 48, 72, and 96 hpf, and dead embryos or larvae were removed at every examination.” The authors had just documented the mortality every 24 hours until 96 hpf. At every examination, all dead embryos were removed to prevent contamination of wells. The result tables with mortality scores are provided as supplementary data in excel file “Raw data for REE LC50.xlsx”. In addition, the mortality curve is now presented in the manuscript (Figure 1) to visualize the the LC50 result. To avoid misconception, the authors further remove the sentences in line 118-125 that explained the observation of zebrafish development staging via their morphology since no morphology or malformations were displayed in the manuscript. To further avoid misinterpretation, the title also changed into “Lanthanides Toxicity in Zebrafish Embryos are Correlated to their Atomic Number”. (update in Figure 1)
The results from PMR test are weakly and illogically presented. It is difficult for the reader to understand what exactly resulted from this tests. Maybe it would be a good idea to include some results from supplementary materials into the main text.
The authors thank the reviewer for pointing this out matter. Therefore, the PMR results have been displayed in Figure 2. In here, we provided additional toxicity evaluation by observing the zebrafish movement. It further displayed even at lower concentration 1 ppm, the Lanthanides, both from LREE and HREE group could display alterations in the PMR activity. However, endpoints coming from the PMR test display low Pearson's correlation coefficient to the 96h LC50. (update in Figure 2)
Round 2
Reviewer 3 Report
I would like to thank the Authors for their changes and additions. However, there are still small but important issues that should be corrected/explained/added. Firstly, what is the baseline (0) in the Fig2? Is it continuous light? It has to be explained, because in this form it is not clear to which value is the reference. Secondly, I suggest to add graphs (in the main text) presenting values form S2. The results are interesting and shouldn’t be hidden in supplementary files.
Last thing, it would be good idea to better discuss the opposite effects from LC50 test and PMR test.
Author Response
Comments and Suggestions for Authors
I would like to thank the Authors for their changes and additions. However, there are still small but important issues that should be corrected/explained/added. Firstly, what is the baseline (0) in the Fig2? Is it continuous light? It has to be explained, because in this form it is not clear to which value is the reference. Secondly, I suggest to add graphs (in the main text) presenting values form S2. The results are interesting and shouldn’t be hidden in supplementary files.
The authors thank the reviewer for pointing out this matter. For the PMR result in Figure 2 (now it’s changed to Figure 3), the last and first minute of each photoperiod (three dark and three light cycles) from total distance endpoints are used to calculate PMRs. As described by Steele et al. (2018) in the figure below. Two dark and two light periods of photomotor responses are measured. The last (a, c, e, and g) and first (b, d, f, and h) minute of each photoperiod are used to calculate PMRs. Photomotor responses of zebrafish are measured as the change in mean (±SEM) distance traveled between the last minute of an initial photoperiod and the first minute of the following period.
Steele, W.B.; Mole, R.A.; Brooks, B.W. Experimental protocol for examining behavioral response profiles in larval fish: application to the neuro-stimulant caffeine. JoVE (Journal of Visualized Experiments) 2018, e57938.
In this study, some modifications based on our previous publications were applied to conduct the PMR test. Prior the test, fish were acclimatized for 30 minutes and instead of starting with dark cycle, we started with light cycle first, then followed by the dark cycle at the end of the test. Therefore, in the Figure 2 (changed to Figure 3), the PMR result starts with Dark 1 which indicates the first transition from light to dark cycle followed by Light 1 that indicates the first transition form dark to light cycle, and so on until the end of photoperiod. Additional explanations have been added in the manuscript to clarify this matter (line 268-270).
In addition, we also added graphs as Figure 2 in the main text presenting values from Table S2 as suggested by the reviewer. Additional description regarding the Lanthanide’s effect toward locomotor activity based on three endpoints: total distance, burst, and rotation count was also added in the manuscript (line 249-258).